SCIENCE FORUM

# A Global Immunological Observatory to meet a time of pandemics

**Abstract** SARS-CoV-2 presents an unprecedented international challenge, but it will not be the last such threat. Here, we argue that the world needs to be much better prepared to rapidly detect, define and defeat future pandemics. We propose that a Global Immunological Observatory and associated developments in systems immunology, therapeutics and vaccine design should be at the heart of this enterprise.

**MICHAEL J MINA[†]\*, C JESSICA E METCALF[†]\*, ADRIAN B MCDERMOTT, DANIEL C DOUEK, JEREMY FARRAR AND BRYAN T GRENFELL**

**\*For correspondence:** mmina@
hsph.harvard.edu (MJM);
cmetcalf@princeton.edu (CJEM)

[†]These authors contributed
equally to this work

**Competing interests:** The
authors declare that no
competing interests exist.

**Reviewing editor:** Clifford J
Rosen, Maine Medical Center
Research Institute, United States

Early in a pandemic, clinical surveillance and pathogen sequencing are essential first steps, bolstered by extensive testing to characterize and initiate control of the pandemic trajectory. But this tracking of infection provides only part of the picture (*Figure 1A*). Specifically, it cannot tell us the trajectory of susceptible individuals who, together with cases, determine the shape and size of the pandemic; the other major group, recovered (potentially immune) individuals, who drive the dynamics of herd immunity, is also not directly observable. Serological surveys can characterize these key hidden variables; however, validation and interpretation are difficult issues (especially for novel pathogens, as SARS-CoV-2 currently illustrates). Furthermore, extensive universal and frequent population sampling for serology is not part of the routine surveillance armory. A Global Immunological Observatory (GIO) would address all of these gaps (*Metcalf et al., 2017*; *Metcalf et al., 2016*).

In theory, we have tremendous ability to deploy multiplex testing for immune responses to pathogens (using techniques ranging from classic ELISAs to phage display approaches) as well as pathogen presence (via genetic sequencing, antigen detection and so on). Further, these methods work with an increasing variety of accessible sample types (saliva, blood spots and so on) which are minimally invasive. Yet, despite decades of technical progress in measurement of both immune responses and pathogen presence (including SARS-CoV and MERS-CoV), which together reveal the core processes driving pathogen transmission (*Figure 1A*), the global health community was unable to identify and model local circulation of SARS-CoV-2 in a timely fashion in almost any setting.

How could we better deploy and refine tools for sophisticated pathogen surveillance to better meet likely future comparable threats (*Metcalf et al., 2016*)? Several problems have prevented this in the current pandemic. First, a shortage of testing capacity and a paucity of historical and contemporary samples to ground analyses combined to cripple our inferential capacity. More fundamentally, despite huge recent progress in immunology, the complexity of the immune system remains a barrier: a revolution in the infrastructure of immune surveillance and systems immunology to generate new understanding and resultant techniques is required. A number of innovations are in reach to step away from the status quo by building GIO, structured around three core sample types.

1. Routinely collected seasonal and international surveillance samples to define the

baseline (such as clinical discard blood specimens from adults, or blood bank and plasma donor samples, representative random sampling and so on) and thus capture anomalies reflecting immune responses to emerging threats. Ideally, this would proceed in parallel with extensive pathogen detection and sequencing, see below.

2. Repeated samples from cohorts (ideally across the full age range, and including birth cohorts) to characterize the mechanisms underlying the ontogeny and time-course of immunity. This would be invaluable in the current crisis for teasing out immune correlates of protection.

3. To anticipate zoonotic threats, a multi-species extension of GIO replicating these surveillance streams in key reservoir species (notably bats), and associated at-risk occupations, is an important extension (*Daszak et al., 2020*).

Such samples are a necessary condition for GIO, however, they will not, in themselves, be sufficient – a series of technological developments are also required: to define the core endemic pathogen imprint on the individual and thus population level immune function, necessary to enable identification of departures from it. Traditionally, ELISAs are the foundation of public health immunological surveillance, although they largely remain limited in throughput, both in the numbers of specimens tested and numbers of pathogen-specific antibodies detected. Advances in highly multiplexed, comprehensive serological evaluations of known and potential pathogen exposure (e.g., microarray chips, VirScan [*Xu et al., 2015*; *Khan et al., 2020*]) are increasingly available. Simultaneous epitope and T and B cell repertoire identification such as T-scan (*Kula et al., 2019*), coupled with direct pathogen detection will provide a much more exhaustive picture than currently available of pathogen exposure and immune response.

Critically, deployment of the observatory would have allowed us to rapidly achieve a preliminary understanding of the dynamics of immunity – even a few weeks gained would have been beneficial in the current crisis. For example, with GIO operational, ongoing evaluation of clinical discard or blood donor specimens (e.g., using VirScan) would have enabled fast detection of serological shifts away from baseline. In particular, we would have readily detected excess cross-reactive antibodies for seasonal coronaviruses, SARS, MERS or any of an additional number of animal coronaviruses already present in the VirScan library. The value of routine measurements of such a broad diversity of antibodies (e.g. VirScan detects antibodies to hundreds of thousands of potential pathogen epitopes with only a microliter of blood) is high sensitivity to perturbations in the antibody repertoires including from zoonotic pathogens not known to infect humans. Through GIO, routine evaluation of international longitudinal sample sets would have been available to guide our understanding of attack-rate in vulnerable populations and in children, sero-prevalence, acquisition of sero-positivity and how they map to immunity, potentially vital weeks earlier. Further, even before the outbreak, many of the fundamental immunological uncertainties revealed by the current pandemic could have been addressed in 'peacetime': mass sampling coupled with technical refinements in measurements, and immuno-epidemiological modeling would have shed important light on the potential for and magnitude of cross-reactivity to existing coronaviruses, the role of cellular vs. humoral immunity, the duration of immunity, and the potential for immune pathogenesis from autoimmunity to antibody dependent enhancement.

Beyond immune-based detection, ideally GIO would exist next to a similar Global Pathogen Observatory (GPO) for a parallel effort surveilling and sequencing pathogens themselves, potentially informed by the immunological context GIO reveals. In particular, GPO could build on seminal previous efforts in pathogen discovery (eg, PREDICT, the Human Virome Project), rooted in the disease ecosystem perspective of OneHealth (*Daszak et al., 2020*; *Carroll et al., 2018*).

## From detection to control

Of course, we must not simply observe a pandemic as it evolves. We must act in parallel and we must translate our knowledge into directly reducing morbidity and mortality in all vulnerable populations. GIO and GPO provide a unique lens for achieving this. The necessary technologies fundamentally already exist. The time from release of the SARS-CoV-2 sequence to a needle in an arm in the first Phase one vaccine trial was only 65 days (https://www.clinicaltrials.gov/ct2/show/NCT04283461). This is a remarkable

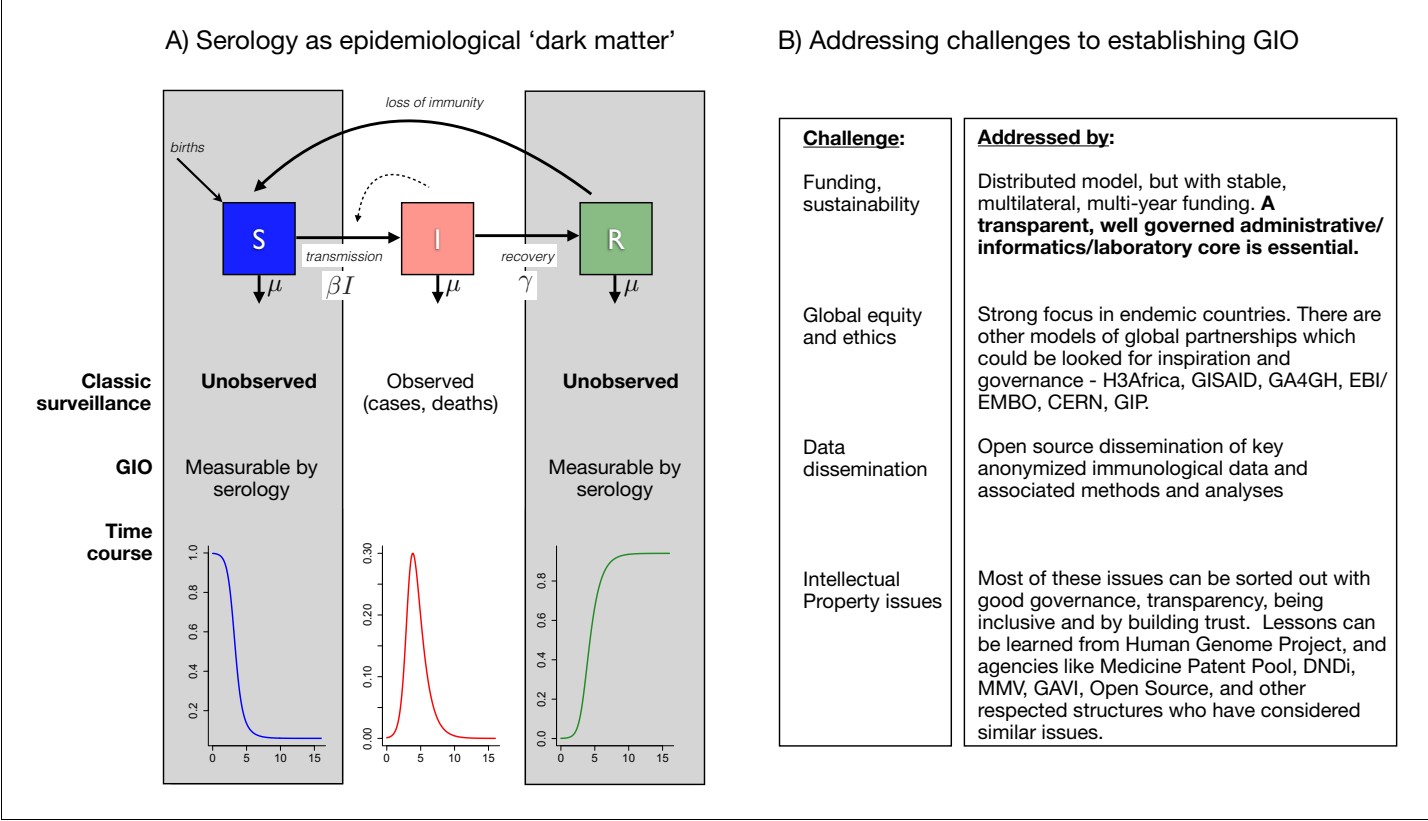

**Figure 1.** The goals of a Global Immunological Observatory, and the challenges involved in establishing such a body. (**A**) The epidemiological process (at its simplest) can be captured as a set of flows from susceptibles (S) to infected individuals (I), which occurs at a rate defined by the numbers of infected individuals and the rate at which they encounter susceptible individuals (a function of human behavior) and then successfully transmit to them – these last two processes are here captured by the parameter $\beta$. Infected individuals may then recover (entering the R class), and may or may not then become susceptible again. Typical surveillance only captures the I class: innovations around a Global Immunological Observatory (GIO) would provide a window onto the 'dark matter' of epidemiology (that is, the S and R classes). (**B**) Establishing a GIO will involve addressing challenges related to funding and sustainability, global equity and ethics, data dissemination, and intellectual property.

achievement made possible through the recent development of easily manufactured, implementable and deployable vaccine platforms, such as mRNA or DNA constructs.

Contemporaneously, permissive animal challenge models should be established to test the vaccine and also to aid in the identification of neutralizing monoclonal antibodies which could be used in short term prevention and as therapeutic agents. This endeavor should go hand in hand with identification of neutralizing monoclonal antibodies in people who have recovered from the infection as well as with detailed analysis of their T cell responses. Finally, and this all flows logically from the initial serological signal and identification of the pathogen's sequence and in vitro production of its proteins, the establishment of assays to measure the breadth, affinity, receptor binding competition analyses and neutralizing capacity of specific antibodies

would be a critical part of this process. Importantly, all these activities can and should be done in parallel. Indeed, with 'peacetime' sampling of reservoir species by GIO and GPO, a library of mRNA/DNA candidates could be generated preemptively. More broadly, GIO could provide considerable insights into other immune-related diseases: from cancer to autoimmune disorders.

We argue that a Global Immunological Observatory is essential for understanding and combating future pandemics. The GIO/GPO infrastructure would bootstrap our ability to engage with a series of fundamental questions associated with the other key processes in pandemics, such as the transmission process – encompassing the biophysics of pathogen spread between individuals, as well as the role of patterns of social mixing. To make full use of immunological and pathogen observatories,

cross fertilization between many fields will be required: spanning systems immunology, virology and physical sciences, through to public health and social science. Major progress in computational biology and disease modeling would also be enabled by these rich and varied datasets. In particular, a key gap in pandemic and endemic disease understanding is the cross-scale phylodynamic interaction between viral evolution and immune kinetics in individuals and nonlinear epidemic dynamics at population and global scales (*Grenfell et al., 2004*). Establishment of GIO/GPO would generate a step change in knowledge at this key interface.

Many lessons will emerge from this pandemic, but we need to learn even more before the next inevitable outbreaks occur. GIO and a larger pathogen enterprise will require significant annual investments, but this will be trivial compared to the costs of future pandemics. Establishing GIO/GPO would require solving a range of important practical issues, from sustainable funding, governance, global equity, intellectual property and relationships with industry, but there are existing models and clear paths to addressing all of these challenges (*Figure 1B*). Lessons for infrastructure, funding and governance could also be learned from the virological and immune phenotypes collected to inform strain selection for seasonal influenza vaccination coordinated by the WHO (*Morris et al., 2018*). A variety of distributed models are possible for GIO/GPO, but an essential condition is the transparent and effective governance of administrative, informatics and laboratory cores.

Another field where understanding of spatiotemporal dynamics is crucial to addressing a deeply applied problem is weather and climate science. The skill of weather forecasts was dramatically improved by deployment of buoys measuring Sea Surface Temperature across the oceans of the world (http://www.argo.ucsd.edu/About_Argo.html). Similarly, in the context of global health, we have the means to deploy measurements crucial both to pandemic preparedness, and endemic disease control, using the delicate and responsive sensor that is all of our immune systems (and the immune systems of all the reservoir species). And as this pandemic has repeatedly shown, early warning and rapid response can make dramatic differences that translate directly to immensely favorable outcomes. Susceptibility and immunity have been the 'dark matter' of epidemic dynamics; GIO could reveal them, to the considerable benefit of global health.

**Michael J Mina** is at the Center for Communicable Disease Dynamics, the Department of Epidemiology and the Department of Immunology and Infectious Diseases, Harvard School of Public Health, and the Department of Pathology, Brigham and Women's Hospital, Boston, United States

mmina@hsph.harvard.edu

https://orcid.org/0000-0002-0674-5762

**C Jessica E Metcalf** in the Department of Ecology and Evolutionary Biology and the Woodrow Wilson School, Princeton University, Princeton, United States

cmetcalf@princeton.edu

https://orcid.org/0000-0003-3166-7521

**Adrian B McDermott** is in the Vaccine Research Center, National Institutes of Health, Bethesda, United States

https://orcid.org/0000-0003-0616-9117

**Daniel C Douek** is in the Vaccine Research Center, National Institutes of Health, Bethesda, United States

**Jeremy Farrar** is at the Wellcome Trust, London, United Kingdom

**Bryan T Grenfell** is in the Department of Ecology and Evolutionary Biology and the Woodrow Wilson School, Princeton University, Princeton, and the Fogarty International Center, National Institutes of Health, Bethesda, United States

https://orcid.org/0000-0003-3227-5909

*Author contributions:* Michael J Mina, Conceptualization, Writing - original draft, Writing - review and editing, Contributed equally with CJEM; C Jessica E Metcalf, Conceptualization, Writing - original draft, Writing - review and editing, Contributed equally with MJM; Adrian B McDermott, Daniel C Douek, Jeremy Farrar, Bryan T Grenfell, Conceptualization, Writing - original draft, Writing - review and editing

*Competing interests:* The authors declare that no competing interests exist.

## Funding

The authors declare that there was no funding for this work.

## Decision letter and Author response

Decision letter https://doi.org/10.7554/eLife.58989.sa1
Author response https://doi.org/10.7554/eLife.58989.sa2

## Additional files

### Data availability

No data is involved in this manuscript.

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
