## [Decision Letter]

Thank you for submitting your article, "A Global Immunological Observatory to meet a time of pandemics" to *eLife*. Your article has been reviewed by two reviewers [reports below] and I am pleased to tell you that it will be accepted for publication as a Feature Article once it has been revised in response to the comments from the referees and a number of editorial points.

To help expedite the revision process I have extracted the points from the reviews that you need to address – please see points 1-5 below.

Points to address from the reviewers

The key question is not should a GIO & GPO be implemented (they should), but, why has progress not been made on implementing such structures since they were originally proposed. What was not discussed were the financial, governance and equity barriers that have prevented the establishment of a GIO.

1) "Finance. Biobanking at scale is expensive - who pays? How is this sustained in the long term? Where are all the -80C freezers? And who is maintaining them?"

- From the Features Editor: Please consider adding a few sentences about these questions, or at least mention that these questions will need to be answered if the GIO project is to move ahead.

2) "Governance. To meet the ambitious vision of the authors, the 'global' part of GIO is especially important. Who owns the samples? Who determines access? How does transfer occur across international borders?"

- From the Features Editor: Again, please consider adding a few sentences about these questions, or at least mention that these questions will need to be answered.

3) "Equity. Most important of all in my opinion. There is a big gap between the low and middle income countries (LMIC) where many emergent pathogens originate, and high income countries with elite institutions and laboratories possessing advanced technologies (e.g. VirScan). Global observatories require global participation, and the perspective of LMIC researchers and public health personnel is missing."

- From the Features Editor: Please add a paragraph about equity barriers.

4) "The proposed GIO would also build on existing infrastructure (GPO, Global Virome project); if space allows, it may be helpful to further expand on this part."

- From the Features Editor: Please add a few sentences about how GIO might build on existing infrastructure.

5) "I really like the analogy of the deployment of buoys for measuring Sea Surface Temperature. Such buoys produce data which is easily shared and reproduced. A GIO would produce serum of finite quantity which must be stored somewhere, and can only be used by a select few.

- From the Features Editor: Please add a few sentences about this issue (unless it is already covered by your answers to the points above).

Reports from the reviewers

Reviewer 1

I enjoyed reading your paper, "A Global Immunological Observatory to meet a time of pandemics." In addition to clinical surveillance and pathogen sequencing, you highlight the following points to managing future pandemics: 1) collection of core samples 2) pairing advanced serologic sampling with technical refinements in measurements and immuno-epidemiological modeling 3) building on serologic sampling to test vaccines and neutralizing monoclonal antibodies.

Core samples would collected through seasonal and international surveillance , repeated sampling through cohorts, and these surveillance streams would be collected in key reservoir species as well. Importantly, the inclusion of longitudinal cohorts internationally would help us better understand the attack-rates of pandemics in vulnerable populations. COVID-19 has exacerbated longstanding inequities in health, and it is an important point to include vulnerable populations in the proposed serologic sampling.

The authors assert that a Global Immunological Observatory (GIO) essential for understanding and combating future pandemics. There are multiple strengths to this article; it is well-written, and it clearly outlines the need for immunological observatory. I have no major concerns.

The proposed GIO would also build on existing infrastructure (GPO, Global Virome project); if space allows, it may be helpful to further expand on this part.

Reviewer #2

With this reviewer, Mina and Metcalf are preaching to the choir. I am in full agreement that it would have been incredibly beneficial to have had an established Global Immunological Observatory (GIO) to respond to the ongoing SARS-CoV-2 pandemic. Indeed, the benefit of such a structure was just as clear four years ago when a subset of the authors proposed essentially the same thing under the name of a World Serum Bank in a Viewpoint in The Lancet (a piece I very much admired). This article revisits the same territory as before but highlights the important benefits that would have been possible had a GIO been implemented. Admittedly, this is shutting the stable door after the horse has bolted. Indeed, this point was made in the original Viewpoint with respect to the 2009 influenza pandemic. Despite this, this point is more valid than ever and deserves to be heard again, and widely discussed in a forum such as *eLife*.

Since this is an opinion piece, I'll brazenly take the opportunity to throw in my 2 cents. As these are personal opinions related to the topic, and not criticisms of the authors' valid points, they can be ignored at the discretion of the editor and authors.

The key question is not should a GIO & GPO be implemented (they should), but, why has progress not been made on implementing such structures since they were originally proposed. What was not discussed were the financial, governance and equity barriers that have prevented the establishment of a GIO.

Finance. Biobanking at scale is expensive - who pays? How is this sustained in the long term? Where are all the -80C freezers? And who is maintaining them?

Governance. To meet the ambitious vision of the authors, the 'global' part of GIO is especially important. Who owns the samples? Who determines access? How does transfer occur across international borders?

Equity. Most important of all in my opinion. There is a big gap between the low and middle income countries (LMIC) where many emergent pathogens originate, and high income countries with elite institutions and laboratories possessing advanced technologies (e.g. VirScan). Global observatories require global participation, and the perspective of LMIC researchers and public health personnel is missing.

I really like the analogy of the deployment of buoys for measuring Sea Surface Temperature. Such buoys produce data which is easily shared and reproduced. A GIO would produce serum of finite quantity which must be stored somewhere, and can only be used by a select few.

---

## [Author Response]

[We repeat the reviewers’ points here in italic, and include our replies point by point, as well as a description of the changes made, in Roman.]

Points to address from the reviewersThe key question is not should a GIO & GPO be implemented (they should), but, why has progress not been made on implementing such structures since they were originally proposed. What was not discussed were the financial, governance and equity barriers that have prevented the establishment of a GIO.

Many thanks for these suggestions - we have altered the manuscript along the lines suggested, as detailed below. The principle change is the extension of Figure 1 to include a second panel that details some of the issues raised; but we have also added some specific points to the text.

1) "Finance. Biobanking at scale is expensive - who pays? How is this sustained in the long term? Where are all the -80C freezers? And who is maintaining them?"- From the Features Editor: Please consider adding a few sentences about these questions, or at least mention that these questions will need to be answered if the GIO project is to move ahead.

In the second panel to the figure (that lists challenges to establishing GIO/GPO and approaches to meeting them), we also emphasize the importance of sustainable funding from multiple sources. However, we don’t think it is helpful to be specific about such sources at this point and in this particular manuscript, so we have not expanded further on this.

2) "Governance. To meet the ambitious vision of the authors, the 'global' part of GIO is especially important. Who owns the samples? Who determines access? How does transfer occur across international borders?"- From the Features Editor: Again, please consider adding a few sentences about these questions, or at least mention that these questions will need to be answered.

Both the mixed funding sources (described above) and the rich collaboration across the research community that we envisage will result in complex ownership of samples. The emergent data (rather than the samples) encompass much of the value and are much more straightforward to share equitably and publicly. We therefore focus on dissemination of anonymized data (rather than samples) for GIO/GPO. We now make this point in Figure 1B.

3) "Equity. Most important of all in my opinion. There is a big gap between the low and middle income countries (LMIC) where many emergent pathogens originate, and high income countries with elite institutions and laboratories possessing advanced technologies (e.g. VirScan). Global observatories require global participation, and the perspective of LMIC researchers and public health personnel is missing."- From the Features Editor: Please add a paragraph about equity barriers.

We point to this in the second panel of Figure 1B, where we note specific models that could be used; and also in the text, where we state that: “Establishing GIO/GPO would require solving a range of important practical issues, from sustainable funding, governance, global equity, intellectual property and relationships with industry, but there are existing models and clear paths to addressing all of these challenges.”

4) "The proposed GIO would also build on existing infrastructure (GPO, Global Virome project); if space allows, it may be helpful to further expand on this part."- From the Features Editor: Please add a few sentences about how GIO might build on existing infrastructure.

We have included a paragraph towards the end of the text, where we state: “Many lessons will emerge from this pandemic- but we need to learn even more before the next inevitable outbreaks occur. GIO and a larger pathogen enterprise will be significant annual investments - but trivial compared to the costs of future pandemics. Establishing GIO/GPO would require solving a range of important practical issues, from sustainable funding, governance, global equity, intellectual property and relationships with industry, but there are existing models and clear paths to addressing all of these challenges (Figure 1B). Lessons for infrastructure, funding and governance could also be learned from the virological and immune phenotypes collected to inform strain selection for seasonal influenza vaccination coordinated by the WHO (Morris et al. 2018). A variety of distributed models are possible for GIO/GPO, but an essential condition is the transparent and effective governance of administrative, informatics and laboratory cores.”

5) "I really like the analogy of the deployment of buoys for measuring Sea Surface Temperature. Such buoys produce data which is easily shared and reproduced. A GIO would produce serum of finite quantity which must be stored somewhere, and can only be used by a select few.- From the Features Editor: Please add a few sentences about this issue (unless it is already covered by your answers to the points above).

This is an excellent point. To address this to some extent, we now focus on public and equitable sharing of the data, rather than the samples, as sharing these is more clearly straightforwardly actionable.

Reports from the reviewersReviewer 1I enjoyed reading your paper, "A Global Immunological Observatory to meet a time of pandemics." In addition to clinical surveillance and pathogen sequencing, you highlight the following points to managing future pandemics: 1) collection of core samples 2) pairing advanced serologic sampling with technical refinements in measurements and immuno-epidemiological modeling 3) building on serologic sampling to test vaccines and neutralizing monoclonal antibodies.Core samples would collected through seasonal and international surveillance , repeated sampling through cohorts, and these surveillance streams would be collected in key reservoir species as well. Importantly, the inclusion of longitudinal cohorts internationally would help us better understand the attack-rates of pandemics in vulnerable populations. COVID-19 has exacerbated longstanding inequities in health, and it is an important point to include vulnerable populations in the proposed serologic sampling.The authors assert that a Global Immunological Observatory (GIO) essential for understanding and combating future pandemics. There are multiple strengths to this article; it is well-written, and it clearly outlines the need for immunological observatory. I have no major concerns.The proposed GIO would also build on existing infrastructure (GPO, Global Virome project); if space allows, it may be helpful to further expand on this part.

Great suggestion - as mentioned above, we now point to some of these prior examples explicitly, stating: “In particular, GPO could build on seminal previous efforts in pathogen discovery (eg, PREDICT, the Human Virome Project), rooted in the disease ecosystem perspective rooted in OneHealth (Daszak, Olival, and Li 2020; Carroll et al. 2018).” And also: “Lessons for infrastructure, funding and governance could also be learned from the virological and immune phenotypes collected to inform strain selection for seasonal influenza vaccination coordinated by the WHO (Morris et al. 2018). A variety of distributed models are possible for GIO/GPO, but an essential condition is the transparent and effective governance of administrative, informatics and laboratory cores.

Reviewer #2:With this reviewer, Mina and Metcalf are preaching to the choir. I am in full agreement that it would have been incredibly beneficial to have had an established Global Immunological Observatory (GIO) to respond to the ongoing SARS-CoV-2 pandemic. Indeed, the benefit of such a structure was just as clear four years ago when a subset of the authors proposed essentially the same thing under the name of a World Serum Bank in a Viewpoint in The Lancet (a piece I very much admired). This article revisits the same territory as before but highlights the important benefits that would have been possible had a GIO been implemented. Admittedly, this is shutting the stable door after the horse has bolted. Indeed, this point was made in the original Viewpoint with respect to the 2009 influenza pandemic. Despite this, this point is more valid than ever and deserves to be heard again, and widely discussed in a forum such as eLife.

Many thanks!

Since this is an opinion piece, I'll brazenly take the opportunity to throw in my 2 cents. As these are personal opinions related to the topic, and not criticisms of the authors' valid points, they can be ignored at the discretion of the editor and authors.The key question is not should a GIO & GPO be implemented (they should), but, why has progress not been made on implementing such structures since they were originally proposed. What was not discussed were the financial, governance and equity barriers that have prevented the establishment of a GIO.

We take this point, and have tried to address it in the current version, specifically expanding on the practicalities in both Figure 1B, and in the text (as described above). However, we have stopped short of outlining the full scope of how this might work in detail –this would be a much longer set of documents, and is perhaps beyond the scope of what is achievable here. In terms of why so little progress towards GIO has been made so far, we agree whole-heartedly with your sentiments, but think that it is important to look forward rather than back at this point.

Finance. Biobanking at scale is expensive - who pays? How is this sustained in the long term? Where are all the -80C freezers? And who is maintaining them?

We now mention this in Figure 1B, and further outline our thinking above. Although we agree that sustainable funding will be critical, we don’t think it is helpful to be specific about the funding such sources at this point and in this particular manuscript, so we have not expanded further on this. We state that: “Establishing GIO/GPO would require solving a range of important practical issues, from sustainable funding, governance, global equity, intellectual property and relationships with industry, but there are existing models and clear paths to addressing all of these challenges (Figure 1B).”

Governance. To meet the ambitious vision of the authors, the 'global' part of GIO is especially important. Who owns the samples? Who determines access? How does transfer occur across international borders?

We strongly agree with this point, and try and reinforce it in both the text (stating that: “Establishing GIO/GPO would require solving a range of important practical issues, from sustainable funding, governance, global equity, intellectual property and relationships with industry, but there are existing models and clear paths to addressing all of these challenges (Figure 1B).”) and in Figure 1B.

Equity. Most important of all in my opinion. There is a big gap between the low and middle income countries (LMIC) where many emergent pathogens originate, and high income countries with elite institutions and laboratories possessing advanced technologies (e.g. VirScan). Global observatories require global participation, and the perspective of LMIC researchers and public health personnel is missing.

We strongly agree with this point, as articulated above.

I really like the analogy of the deployment of buoys for measuring Sea Surface Temperature. Such buoys produce data which is easily shared and reproduced. A GIO would produce serum of finite quantity which must be stored somewhere, and can only be used by a select few.

This is a great point: clearly, the data from GIO/GPO could never be quite as timely as that emanating from the buoy network; the samples, in particular, are not easily shared. Accordingly, in this version, we focus on provision of anonymized data, which should be feasible in a timely fashion assuming requisite technical and informatic advances.